# Feasibility and Preliminary Efficacy of Wearable Focal Vibration Therapy on Gait and Mobility in People with Multiple Sclerosis: A Pilot Study

**DOI:** 10.3390/bioengineering12090932

**Published:** 2025-08-29

**Authors:** Hongwu Wang, Yun Chan Shin, Nicole J. Tester, Torge Rempe

**Affiliations:** 1Department of Occupational Therapy, College of Public Health and Health Professions, University of Florida, Gainesville, FL 32611, USA; yunchan.shin@phhp.ufl.edu; 2University of Florida Health Rehabilitation at the Norman Fixel Institute for Neurological Disorders, University of Florida, Gainesville, FL 32628, USA; nicole.tester@ufhealth.org; 3Department of Neurology, College of Medicine, University of Florida, Gainesville, FL 32610, USA; torge.rempe@neurology.ufl.edu

**Keywords:** focal vibration therapy, multiple sclerosis, motion capture analysis, gait biomechanics, feasibility, wearable

## Abstract

Multiple sclerosis (MS) is a chronic disease of the central nervous system that significantly impairs gait and mobility, contributing to a high risk of falls, reduced participation in daily activities, and diminished quality of life. Despite existing interventions such as exercise programs and pharmacological treatments, challenges such as fatigue, pain, and limited accessibility underscore the need for alternative therapies. Focal vibration therapy (FVT) has shown promise in improving gait, reducing spasticity, and enhancing mobility in people with MS (PwMS). However, further research is required to evaluate its long-term feasibility and optimize its parameters. This study examined the feasibility and preliminary efficacy of a home-based four-week wearable FVT device on gait and explored how FVT parameters impact gait and mobility outcomes. In this pilot double-blind randomized controlled trial, 22 PwMS were randomized into control and vibration groups (four FVT groups with varying vibration intensities/durations). Participants wore Myovolt^®^ vibrators on distal quadricep muscles near the rectus femoris insertion (approximately 2 cm from the medial edge of the patella), gastrocnemius/soleus, and tibialis anterior muscles (10 min/muscle, 3 days/week, 4 weeks). Feasibility was evaluated via adherence and satisfaction (QUEST 2.0, interviews). Gait (3D motion analysis) and mobility (T25FW) were assessed at baseline and post-intervention. Data were analyzed using descriptive/inferential statistics and thematic analysis. Of 22 participants, 17 completed post-intervention (16 intervention, 1 control). Wearable FVT showed promising feasibility, with high satisfaction despite minor adjustability issues. Intervention groups improved gait speed (*p* = 0.014), stride length (*p* = 0.004), and ankle angle (*p* = 0.043), but T25FW was unchanged (*p* > 0.05). High-intensity FVT enhanced knee/hip moments. This study’s results support the feasibility of wearable FVT for home-based management of mobility symptoms in MS with high participant satisfaction and acceptance. Notable gains in gait parameters suggest FVT’s potential to enhance neuromuscular control and proprioception but may be insufficient to lead to mobility improvements. Subgroup analyses highlighted the impact of vibration intensity and duration on knee joint mechanics, emphasizing the need for personalized dosing strategies. Challenges included participant retention in the control group and burdensome biomechanical assessments, which will be addressed in future studies through improved sham devices and a larger sample size.

## 1. Introduction

Multiple sclerosis (MS) is a chronic autoimmune disease of the central nervous system (CNS) characterized by demyelination and axonal damage, leading to disrupted neural signal conduction and a range of neurological symptoms [1,2,3]. Among the most debilitating symptoms are gait and mobility impairments, driven by muscle weakness, spasticity, impaired coordination, sensory disturbances, and fatigue [4,5,6]. These challenges affect 50–70% of people with MS (PwMS), with 30% reporting multiple falls and related injuries within a few months [7,8], significantly reducing physical safety, daily activity participation, and quality of life [9,10].

Current interventions for MS-related mobility impairments, such as pharmacological treatments (e.g., dalfampridine) and exercise programs, face limitations, including side effects, limited efficacy, and accessibility barriers [11,12,13,14]. Fatigue and pain further hinder participation in exercise-based interventions, particularly for those with severe mobility limitations [15]. Consequently, there is a pressing need for safe, accessible, and cost-effective non-pharmacological therapies.

Focal vibration therapy (FVT), a non-invasive and portable intervention, stimulates sensory receptors in muscles and tendons to enhance proprioceptive input, neuromuscular control, and muscle activation [16,17]. FVT has shown potential benefits for gait and balance in various populations, including those with neurological conditions [18,19,20]. Four studies have investigated the use of FVT in PwMS. In people with secondary progressive MS who were non-responsive to common antispasmodic medication, repetitive FVT treatment (100 Hz, 0.2–0.5 mm amplitude) was associated with improved gait function by reducing spasticity and pain and improving quality of life [21]. In another study, walking also improved in PwMS (both those with and without spasticity) who received FVT (9000 Hz, < 0.8 N energy) compared with those who received sham vibration therapy [22]. The participants who received FVT improved gait velocity and cadence and took longer steps over a shorter time than the control group. The results reported by Paoloni et al. [23] (120 Hz) and Spina et al. [22] found that fatigue was significantly reduced in the FVT group, though there was no difference between the FVT and control groups. Ayvat et al. reported that FVT (50 Hz and 100 Hz, 1 mm amplitude) had a similar effect on spasticity, a better effect in increasing fascicle length with 50 Hz frequency, and improved stability in PwMS compared to an exercise program [24]. However, current research on FVT in MS is limited, and the optimal parameters, such as vibration frequency, amplitude, duration, and treatment intensity, have yet to be determined. Previous studies that have applied FVT to the lower extremities of people with MS have used different intensities and frequencies of FVT [21,22,23,24]. The measurements and equipment used to measure the effects also varied, making it difficult to assess the feasibility of their long-term usage [21,22,23,24].

This pilot study evaluated the feasibility and preliminary efficacy of a four-week home-based wearable FVT intervention on gait and mobility in PwMS. It aimed to assess the acceptability and feasibility of wearable FVT for home-based application in PwMS and explore how FVT parameters influence gait and mobility outcomes, with the long-term goal of optimizing treatment dosage to manage MS-related lower extremity symptoms effectively. We hypothesized that wearable FVT is feasible and would improve gait and mobility compared to baseline, with effects persisting at least four weeks post-intervention.

## 2. Materials and Methods

### 2.1. Study Design

This study was a pilot double-blind randomized controlled trial. Feasibility of the wearable FVT was assessed by participant acceptance, adherence, and satisfaction; while efficacy was evaluated using gait and functional mobility measures assessed at two time-points: baseline evaluation before the start of the intervention, and post-intervention assessment immediately after the 4-week treatment period. As a pilot study, the target sample size was 30 participants (6 per group, including control), based on feasibility goals rather than a formal power calculation for efficacy, as is common in pilot trials (e.g., to assess recruitment, retention, and preliminary effect sizes). No a priori power calculation was performed for efficacy outcomes due to the exploratory nature.

### 2.2. Participants

Participants with clinically confirmed MS were recruited from the Oklahoma City area and surrounding clinics. The study was approved by the University of Oklahoma Health Sciences Center Institutional Review Board (IRB number 10785). Inclusion criteria were: (1) MS diagnosis defined by the 2017 revised McDonald criteria [25]; (2) age between 18 and 65 years; (3) subjects reporting impaired walking, assessed by clinical examination; (4) no clinical relapse in past 6 months; (5) ability to understand English instructions; (6) ability to sign an informed consent; and (7) normal or corrected vision. Participants with other symptoms or signs suggestive of peripheral neuropathy or other superimposed neurological or psychiatric conditions that could interfere with walking assessment, medical condition preventing the use of the vibration device, pregnancy, and a clinical diagnosis of moderate or severe dementia (defined as Montreal Cognitive Assessment [26] (MoCA) < 24 at screening) were excluded. Using a computer-generated randomization program, participants were randomized into five groups: a control group receiving non-functional sham vibration (showing the same vibration sounds but no actual vibration) and four intervention groups receiving different vibration dosages, as specified below. Allocation was concealed using sequentially numbered, opaque envelopes prepared by an independent researcher not involved in assessments. Envelopes were opened only after baseline consent. The sham vibration had an identical appearance to the experimental devices, with only auditory sounds, but no vibration. Participants were blinded via an identical device appearance (sham produced sound but no vibration). Outcome assessors (trained students) were blinded to allocation. Data analysts were blinded until the final analysis.

### 2.3. Intervention

Participants allocated to the FVT groups wore a portable vibration device (Myovolt^®^, Christchurch, New Zealand) placed bilaterally on three specific lower extremity muscle groups: the distal quadriceps muscle/tendon region near the rectus femoris insertion (approximately 2 cm from the medial edge of the patella), the gastrocnemius/soleus muscle/tendon region, and the tibialis anterior muscle belly (Figure 1). These muscles were selected because they are directly involved during walking, as supported by previous studies [21,22,23,24]. During each FVT session, participants sequentially received the prescribed focal vibration on the distal quadricep muscles, gastrocnemius, and tibialis anterior muscle groups. For the intervention groups, four vibration dosages were selected: Low-intensity short duration (LISD): 60 Hz frequency, 0.5 mm amplitude, 10 min per muscle (30 min/session); Low-intensity long duration (LILD): 60 Hz, 0.5 mm, two 10-min sessions per muscle with a 1-min break (60 min/session); High-intensity short duration (HISD): 120 Hz, 1 mm, 10 min per muscle (30 min/session); High-intensity long duration (HILD): 120 Hz, 1 mm, two 10-min sessions per muscle (60 min/session). These parameters were selected based on prior studies [21,22,23,24] and our pilot testing with people with strokes and diabetes and manufacturer (Myovolt^®^) recommendations for safe home use. All participants in the intervention groups performed three sessions per week over four consecutive weeks. The vibration intensity (frequency and amplitude) was predetermined for each intervention arm and remained constant throughout the intervention period. Participants in the control group wore the Myovolt device in the same locations using the sham vibrators.

At the initial visit, research staff trained all participants to don and doff the Myovolt device properly to avoid device slippage. All participants learned to switch the device on/off and charge the device. We provided each participant with a simple, intuitive user manual (as shown in the Appendix A) to ensure the correct application at home. Participants were asked to note the date and time of the application of the FVT. Throughout the intervention, we maintained regular contact with participants to monitor adherence, troubleshoot device issues, and answer any questions.

### 2.4. Outcome Measurements

At each assessment, evaluators who did not know the intervention assignment recorded the outcomes. The evaluators were physical or occupational therapy students sufficiently trained by an experienced physical therapy researcher to ensure fidelity.

In the baseline assessment, participants’ age, Body Mass Index (BMI), ethnicity, education level, duration of MS, Expanded Disability Status Scale (EDSS) score, ankle-foot orthosis/orthoses usage, walking aid (s) usage, MS medication, and self-reported physical limitation were collected via study personnel trained by a neurologist. The EDSS constitutes one of the oldest and probably most widely utilized assessment instruments in MS [27]. It has been used in virtually every major clinical trial conducted in MS during the last four decades and in numerous other clinical studies. The EDSS is an ordinal clinical rating scale ranging from 0 (normal neurologic examination) to 10 (death due to MS) in half-point increments. It has been shown to be a valid tool [28] for patients with multiple sclerosis with moderate intra-rater reliability [29].

The feasibility of the FVT intervention was assessed by the participants’ adherence, acceptance, and satisfaction indicated by the Quebec User Evaluation of Satisfaction with Assistive Technology (QUEST 2.0) and semi-structured interviews. The Quebec User Evaluation of Satisfaction with QUEST is a standardized assessment tool designed to evaluate user satisfaction with assistive devices [30,31]. It has been shown with good test–retest reliability in people with MS [32]. Semi-structured interviews were conducted to collect qualitative data from participants about their acceptance, perceived usefulness, and experience of the FVT intervention.

Mobility was quantified using the Timed 25-Foot Walk Test (T25FW) at baseline and post-intervention. T25FW is a mobility test based on a timed 25-foot walk [33], with good test–retest reliability in people with MS [34]. The participant is directed to one end of a marked 25-foot course and is instructed to walk 25 feet as quickly as possible, but safely. The time is calculated from the initiation of the instruction to the start and ends when the participant has reached the 25-foot mark. The task is immediately administered again by having the participant walk back the same distance. Participants may use assistive devices when performing this task.

Quantitative gait parameters were collected using the Qualysis™ (Qualisys AB, Göteborg, Sweden) motion capture system and AMTI™ (Advanced Mechanical Technology, Inc., Watertown, MA, USA) force plates, placing reflective markers on 58 anatomical landmarks (Figure 2), using the Qualysis™ motion capture system’s 12 cameras (sampling rate 120 Hz) and the AMTI™ force plates (sampling rate 1 kHz) to capture gait performance data. A physical therapist with over 15 years of experience in gait biomechanics and 8 years with the Qualisys motion capture system placed the markers. Each participant first performed a static trial to develop their individual skeletal model, which the investigators then exported to Visual3D© (C-Motion, Germantown, MD, USA) for data analysis. Participants then walked at a self-selected speed of 25 feet for three to five trials, using the normal pace they used for everyday life. From these trials, we analyzed and averaged data across all valid gait cycles captured, which typically ranged from 3 to 10 cycles per participant per assessment, depending on the quality of force plate contacts and the individual’s stride length (ensuring clean, artifact-free data). Invalid cycles (e.g., those with partial force plate strikes or marker occlusions) were excluded. Gait cycles were primarily determined using force plate data, where heel strike (initial contact) and toe-off events were identified based on a vertical ground reaction force threshold of 20 N. When force plate data was unavailable or incomplete for a given step (e.g., due to the participant missing the plate), kinematic data from reflective markers (specifically, heel and toe marker trajectories and velocities) were used as a supplement to detect events via Visual3D’s automated pipeline. This hybrid approach ensured accurate event detection while maximizing usable data. Investigators then calculated kinematic and kinetic data using Visual3D©. The following gait parameters were calculated: gait speed, stride length, stride width, stride time, left and right stance time, left and right swing time, cadence, duration of double support, left and right peak knee flexion, left and right peak dorsiflexion, left and right peak plantarflexion, left and right peak knee flexor moment, left and right peak plantar flexor moment, and left and right peak ankle joint power. Visual3D© defines the spatiotemporal measures as follows [35]: gait speed is the stride length divided by stride time, stride length is the distance from heel strike to heel strike on the same side, stride time is the duration of the gait cycle from heel strike to heel strike on the same side, left stance time is the time from left heel strike to left toe off, right stance time is the time from right heel strike to right toe off, left swing time is the time from left toe off to left heel strike, right swing time is the time from right toe off to right heel strike, mean cadence is the average number of steps taken per minute, and duration of double support is the amount of time spent with both feet in contact with the surface.

### 2.5. Statistical Analysis

All data were recorded on standardized case report forms and securely stored for subsequent analysis. All statistical analyses were conducted using IBM^®^ SPSS Statistics for Windows, Version 26.0, and RStudio, Version 2022. Data were first screened for normality and homogeneity of variance before conducting inferential tests. Descriptive statistics were computed for participant demographics, feasibility data. Paired t-tests or Wilcoxon signed-rank tests were used to compare the gait variables and T25FW scores between baseline and post-intervention. Statistical significance was set at *p* < 0.05. Due to the exploratory nature of this pilot study, no adjustments were made for multiple comparisons in the analysis of gait parameters, which may increase the risk of false positives, as noted as a limitation. Effect sizes were calculated using Cohen’s d for paired t-tests, where d = differences in the post and pre means / standard deviation of the difference (the standard deviation of the paired differences). For Wilcoxon signed-rank tests, the matched-pairs rank-biserial correlation (r) was computed and converted to d for consistency. Effect sizes were interpreted as small (d = 0.2–0.49), medium (d = 0.5–0.79), or large (d ≥ 0.8) [36]. Qualitative data from semi-structured interviews were analyzed using thematic analysis to identify key themes related to participant satisfaction and usability.

## 3. Results

A total of 22 participants were recruited, with 19 completing the baseline assessment (17 intervention, 2 control) and 17 completing the post-intervention assessment (16 intervention, 1 control) (Figure 3). Three participants assigned to the control group withdrew from the study before the intervention started, and a baseline assessment due to the lack of perceived usefulness of the sham vibration. Two dropouts occurred, one from the HISD group and one from the control group. Consequently, only one participant from the control group completed the post-intervention assessment. The baseline characteristics of the total participants and each group are described in Table 1 and Table 2. The mean age of the total experimental group, which completed the baseline evaluation, was 49.65 years (SD = 9.59), with 15 out of 17 participants (78.95%) being female. The control group had a mean age of 49.50 years (SD = 9.19), with 1 out of 2 participants being female. Therefore, due to the limited sample size, the initial plan to compare each of the four experimental groups with the control group was limited, and the analysis was conducted on the total experimental group and the data from the four individual experimental groups.

### 3.1. Feasibility of the Wearable FVT Intervention

Seventeen participants completed the QUEST satisfaction survey regarding their experience with FVT (Table 3). Overall satisfaction was high. However, the lowest scores for “ease of adjusting” and “durability” were 2 points, with two participants expressing dissatisfaction with them. Additionally, one, one, and three participants scored “safe and secure,” “comfort,” and “how effective” as 3 points, respectively, reflecting a neutral level of satisfaction.

The feasibility measures are shown in Table 4. Sixteen participants, including one from the control group, completed the semi-structured interviews. Fifteen participants expressed acceptance of the FVT, except the one from the control group, who reported no perceived effect of the FVT. Of the 14 participants who responded that they would like to continue to use the focal vibration device, 9 expressed a desire to purchase it, and 3 indicated that they would consider purchasing it if certain improvements were made. Of the 16 participants, 13 did not experience any pain from using the device, with 2 reporting that the device may have caused muscle spasms or pain. The preference for the device included most participants stating that the device was “easy to use”. Participant feedback (Table 4) indicated overall satisfaction with the FVT intervention but highlighted concerns regarding device adjustability (e.g., strap fit), durability (e.g., wear and tear), and comfort during prolonged use, as reported in semi-structured interviews. These findings suggest areas for device improvement in future iterations.

### 3.2. Gait Parameters

The experimental group demonstrated a significant gait speed increase (*p* = 0.014), along with stride length (*p* = 0.004) and the peak plantarflexion during the pre-swing phase of the more involved side (*p* = 0.043) after the FVT intervention (Table 5 and Table 6). There were no significant changes in other spatiotemporal, kinematic, or kinetic variables (Table 5, Table 6 and Table 7). Trends toward significance were observed in the peak plantarflexion before heel strike (*p* = 0.056), peak hip flexion during stance (*p* = 0.08), peak ankle power in late stance (*p* = 0.086), peak ankle power in early stance (*p* = 0.077), peak knee flexor moment in early swing (*p* = 0.076), and peak knee power absorption in swing phase (*p* = 0.065) of the more involved side, respectively.

For the low intensity subgroup, the peak hip extension angle during the stance phase, the knee extension angle at the heel strike, and the peak knee flexion angle at the stance phase of the more involved side were significantly improved after the intervention. No significant changes were noted for other subgroups. We observed significantly improved peak knee flexion angle during the stance phase, peak knee flexion angle during the swing phase, and peak knee flexion moment during the stance phase in the more involved side in the participants receiving low-intensity FVT. For those who received high-intensity FVT, significantly improved peak hip extension power during the swing phase, peak knee flexion moment during the swing phase, and peak knee power during the stance phase were observed in the less involved side. The peak knee extension moment at the stance phase on the less involved side was the only gait parameter that showed a statistically significant difference when comparing the low-intensity and high-intensity groups.

### 3.3. Mobility

There were no statistically significant (*p* = 0.54) changes in the T25FW scores between the pre- (7.80 ± 3.50 s) and post-assessments (7.68 ± 3.09) for the experimental group. Minimal detectable clinical differences (20% change in the T25FW score [37]) were also not found in the total and each subgroup.

## 4. Discussion

This pilot study addressed a key gap in the literature: while focal vibration therapy (FVT) has shown preliminary benefits for gait and spasticity in people with multiple sclerosis (PwMS) in short-term, clinic-based settings [21,22,23,24], there is limited evidence on the feasibility and efficacy of wearable, home-based FVT over extended periods (e.g., 4 weeks), including optimal dosing parameters (intensity and duration) and their impact on detailed biomechanical gait outcomes. This research was needed because current interventions for MS-related mobility impairments, such as pharmacological treatments and exercise programs, often face limitations like side effects, accessibility barriers, and fatigue-related non-adherence [11,12,15], underscoring the importance of developing safe, portable, non-invasive alternatives to enhance neuromuscular control and daily function in PwMS. The primary purpose of this study was to evaluate the feasibility (adherence, satisfaction, acceptability) and preliminary efficacy of a 4-week home-based wearable FVT intervention on gait biomechanics and mobility in PwMS, while exploring dose–response effects to inform personalized protocols. Our hypothesis, that wearable FVT would be feasible and improve gait and mobility compared to baseline, with effects persisting at least four weeks post-intervention, was partially supported: feasibility was high with notable gait improvements (e.g., speed, stride length), but mobility (T25FW) remained unchanged, likely due to the short intervention duration and the test’s limited sensitivity to subtle biomechanical changes; persistence could not be fully assessed as assessments were conducted immediately post-intervention, though gains were evident at that time-point.

Based on the QUEST satisfaction survey, most participants found the device acceptable and “easy to use.” Semi-structured interviews further highlighted the general acceptance of FVT, with 15 out of 16 participants expressing positive attitudes toward its use. Notably, 14 participants indicated a willingness to continue using the device, with 9 expressing interest in purchasing it and 3 considering purchasing it if improvements were made, i.e., the strap. More than half of the participants expressed a desire for increased vibration. Two participants reported applying vibration to their wrists, noting improved hand function. Importantly, most participants did not report pain related to device use, though two individuals noted minor muscle spasms or pain. Areas for improvement were identified, particularly regarding “ease of adjusting” and “durability,” which received the lowest satisfaction scores. Neutral feedback was also observed for “safety and security,” “comfort,” and “effectiveness.” These concerns resonate with findings from other studies investigating wearable technology for MS and other central neurological disorders, emphasizing the importance of user-centered design to ensure that devices are not only effective but also practical and comfortable for everyday use [38,39,40]. This study supports the promising feasibility of wearable FVT for home-based management of mobility symptoms in MS, with generally positive participant satisfaction and acceptance. However, participant feedback highlighted challenges with device adjustability, durability, and comfort, which must be addressed to enhance usability and widespread adoption.

Improvements in gait parameters, such as increased gait speed, longer stride length, and alterations in ankle angle, aligned with previous studies demonstrating that FVT can positively influence neuromuscular control and proprioceptive feedback [41,42,43]. Similar patterns have been seen in other neurological populations, where kinematic and spatial-temporal gait parameter changes emerge even when mobility measures remain unchanged [44,45]. The lack of significant improvement in the T25FW may be attributed to its focus on short-distance walking, which may not fully capture subtle biomechanical improvements observed in gait parameters. Previous studies [22] suggest that short-distance tests like the T25FW may be less sensitive to neuromuscular changes compared to longer-distance measures, such as the Six-Minute Walk Test. Future studies will incorporate alternative functional mobility measures to better assess the translation of biomechanical gains into clinically meaningful outcomes. Interventions that improve underlying neuromuscular function often require additional time, training, or task-specific practice to translate into clinically meaningful outcomes in everyday mobility [46]. Since the wearable FVT could be self-applied at home, it could be used for a longer duration in future studies to lead to meaningful mobility improvements. This was reflected by the qualitative interview, which showed that more than half of the participants wanted more vibration. No standardized scales, such as the Multiple Sclerosis Impact Scale (MSIS-29), were collected alongside biomechanical measures in this study, as the pilot focused primarily on feasibility and biomechanical outcomes. However, during qualitative interviews, 5 to 6 participants spontaneously reported perceived increases in mobility and energy related to walking, suggesting potential subjective benefits that warrant further investigation. We plan to incorporate validated scales like the MSIS-29 in larger future trials to quantitatively assess these self-reported improvements.

The subgroup analyses indicated that vibration intensity and duration influenced only certain gait parameters, primarily related to knee joint mechanics. These results are consistent with prior research emphasizing the importance of selecting appropriate vibration parameters for different target symptoms [47,48]. The low- and high-intensity groups exhibited different patterns of knee moment changes on the less- and more involved side, which hints at the complexity of tailoring FVT doses and frequencies. These findings underscore the importance of personalized approaches in vibration therapy to optimize efficacy and minimize potential adverse effects. In future studies, we may need to apply different vibration intensities on different sides and adjust the vibration parameters as the rehabilitation progresses. Future studies with larger samples and standardized protocols are warranted to optimize dosing depending on the individual’s baseline impairments, disease severity, and functional goals. We envision combining FVT with strength training (e.g., progressive resistance exercises) or balance training (e.g., Tai Chi), as vibration may enhance proprioception to amplify training effects. Dosing could interact by using lower intensities during acute training to avoid fatigue, or higher intensities for spasticity reduction pre-exercise. This is speculative and requires testing.

There were several limitations to this pilot study. Firstly, this study had a small sample size (n = 17 completing post-intervention assessments, with only one control group participant completing the study), which limited the statistical power and increased the risk of Type I and Type II errors. This study encountered significant challenges in participant recruitment and retention, largely due to the COVID-19 pandemic. Like other research [49], several participants were unable to continue participating in the study or were unable to enroll due to worsening health conditions, reduced clinical visits, and limited transportation. Secondly, only one of the five participants randomized to the control group completed the study, with three declining to participate in the study after trying the sham vibration device, as they did not feel the device would work for them, and one of the two control group participants who completed the baseline evaluation dropped out, making it impossible to compare data between the experimental and control groups. The high dropout rate in the control group, likely due to the lack of perceived benefit from the sham device, compromised the validity of comparisons between intervention and control groups. To address this, a new sham device generating both vibration sound and non-functional low-frequency vibration has been developed for future studies to improve participant retention. A larger follow-up study with an increased sample size and improved control group design is planned to enhance the robustness of findings. Another limitation of this study is that the gait biomechanical analysis was only conducted for the first two visits. One reason for the lack of assessments on the third visit was the assessment burden on the participants. The marker-based motion capture system requires around 40 min of preparation before the walking trail, which many participants felt was burdensome, as many MS participants reported fatigue at the end of the biomechanical test. In addition, participants using walkers or canes reported difficulty completing the biomechanical trail. We have established a new protocol using a markerless motion capture system to reduce the participants’ burdens in future studies. Additionally, the analysis of multiple gait parameters without correction for multiple comparisons may have increased the risk of Type I errors, and these findings should be interpreted cautiously. Future studies will apply appropriate statistical corrections, such as Bonferroni adjustments, to account for multiple comparisons.

## 5. Conclusions

The present results highlight the potential of wearable FVT as a feasible and sustainable treatment for home-based mobility-associated symptoms management among people with MS. High participant satisfaction with wearable FVT supports its integration into MS rehabilitation protocols, promoting patient-centered care and functional independence. Preliminary improvements in gait speed, stride length, and ankle angle suggest FVT’s potential to enhance neuromuscular control, reducing mobility-related disability and fall risk. While these subtle biomechanical improvements did not translate to significant improvements in a short-distance mobility test, they represent an encouraging step toward identifying how FVT can be optimally integrated into rehabilitation programs. Further research should focus on refining FVT parameters, combining vibration with other targeted therapies, and examining outcomes over longer distances and more ecologically valid scenarios to fully realize the potential of FVT in improving the mobility and quality of life of individuals with MS.

## Figures and Tables

**Figure 1 bioengineering-12-00932-f001:**
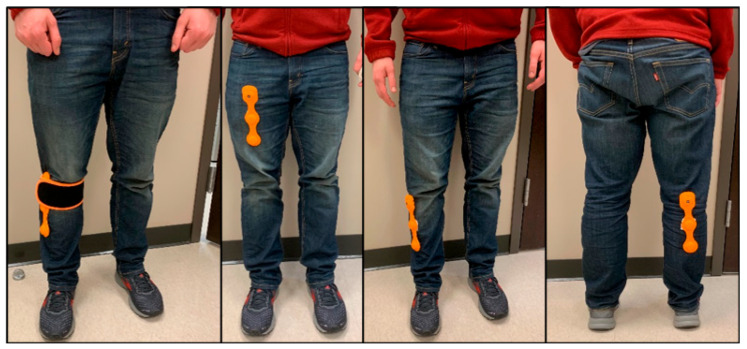
Myovolt wearable FVT device and application sites (left to right: vibration device with strap, quadriceps muscle/tendon region, gastrocnemius/soleus muscle/tendon region, and the tibialis anterior muscle belly).

**Figure 2 bioengineering-12-00932-f002:**
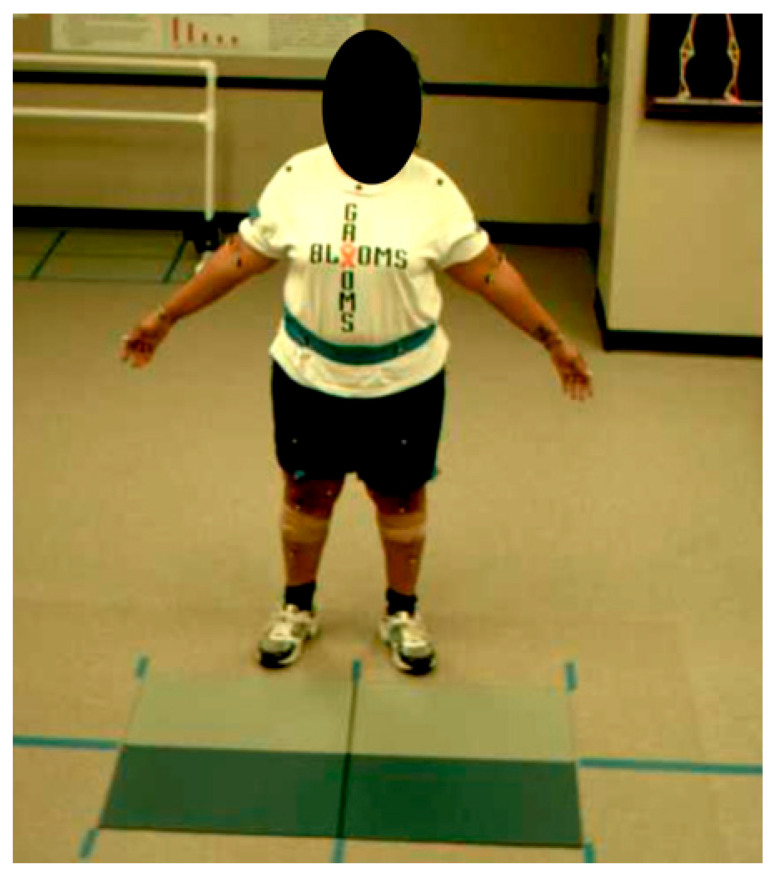
Reflective marker placement.

**Figure 3 bioengineering-12-00932-f003:**
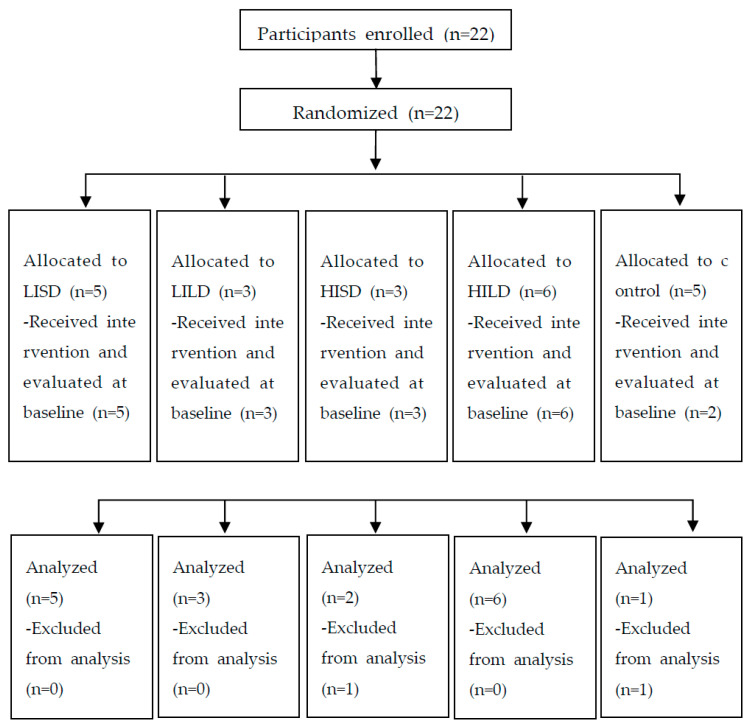
Flow of the trial selection process. Twenty-two participants were enrolled and randomized to different intervention and control groups. Nineteen participants completed the baseline assessment. Seventeen participants completed the post-intervention assessment. LISD: low intensity and short duration; LILD: low intensity and long duration; HISD: high intensity and short duration; HILD: high intensity and long duration.

**Table 1 bioengineering-12-00932-t001:** Participant Characteristics (n = 19).

	Experimental Group (n = 17)	Control Group (n = 2)
Age (mean, (SD))	49.65 (9.59)	49.50 (9.19)
Gender (n, (%)) Male Female	3 (17.65%) 14 (82.35%)	1 (50.00%) 1 (50.00%)
BMI (mean, (SD))	33.31 (7.86)	22.27 (0.97)
Ethnicity (n, (%)) African American Caucasian	6 (35.29%) 11 (64.71%)	1 (50.00%) 1 (50.00%)
Education (n, (%)) GED or High School Diploma Some college Bachelor’s Degree Master’s Degree Doctorate Associate degree Technical Certification	4 (23.53%) 2 (11.76%) 5 (29.41%) 3 (17.65%) 1 (5.88%) 1 (5.88%) 1 (5.88%)	1 (50.00%) 0 (0.00%) 0 (0.00%) 0 (0.00%) 0 (0.00%) 1 (50.00%) 0 (0.00%)
Years of MS (mean, (SD))	9.76 (7.53)	8 (9.90)
Baseline EDSS score (mean, (SD))	3.22 (0.93) ^++^	4.00 (0.00)
Ankle-foot orthoses usage (n, (%)) Yes No	1 (5.88%) 16 (94.12%)	0 (0.00%) 2 (100.00%)
Walking aids usage (n, (%)) Yes No	6 (35.29%) 11 (64.71%)	1 (50.00%) 1 (50.00%)
Using MS medication Yes No	16 (94.12%) 1 (5.88%)	2 (100.00%) 0 (0.00%)
Perceived importance of medication use (mean, (SD))	9.82 (0.53)	9.00 (1.41)
Physical limitations affecting self-care (n, (%)) Problems with hand Problems with feet Vision loss Hearing loss	3 (17.65%) 8 (47.06%) 1 (5.88%) 1 (5.88%)	0 (0.00%) 1 (50.00%) 1 (50.00%) 0 (0.00%)

^++^: n = 16. Includes age, BMI, ethnicity, education level, MS duration, EDSS score, assistive devices usage, use of MS medication, and perceived importance of medication use (1–10 Likert scale: 1 = not important, 10 = very important).

**Table 2 bioengineering-12-00932-t002:** Baseline participants’ demographics by intervention subgroups.

	Low-Intensity Vibration Group (n = 8)	High-Intensity Vibration Group (n = 9)	Short-Duration Vibration Group (n = 8)	Long-Duration Vibration Group (n = 9)
Age (mean, (SD))	50.00 (9.74)	49.33 (10.04)	50.88 (9.63)	48.56 (10.00)
Gender (n, (%)) Male Female	2 (25.00%) 6 (75.00%)	1 (11.11%) 8 (88.89%)	2 (25.00%) 6 (75.00%)	1 (11.11%) 8 (88.89%)
BMI (mean, (SD))	36.24 (7.28)	30.71 (7.82)	34.75 (7.94)	32.03 (8.03)
Ethnicity (n, (%)) African American Caucasian	2 (25.00%) 6 (75.00%)	4 (44.44%) 5 (55.56%)	2 (25.00%) 6 (75.00%)	4 (44.44%) 5 (55.56%)
Education (n, (%)) GED or High School Diploma Some college Bachelor’s Degree Master’s Degree Doctorate Associate degree Technical Certification	1 (12.50%) 2 (25.00%) 3 (37.50%) 1 (12.50%) 1 (12.50%) 0 (0.00%) 0 (0.00%)	3 (33.33%) 0 (0.00%) 2 (22.22%) 2 (22.22%) 0 (0.00%) 1 (11.11%) 1 (11.11%)	1 (12.50%) 1 (12.50%) 3 (32.50%) 2 (25.00%) 1 (12.50%) 0 (0.00%) 0 (0.00%)	3 (33.33%) 1 (11.11%) 2 (22.22%) 1 (11.11%) 0 (0.00%) 1 (11.11%) 1 (11.11%)
Years of MS (mean, (SD))	7.75 (8.48)	11.56 (6.54)	8.13 (7.94)	11.22 (7.29)
Baseline EDSS score (mean, (SD))	2.79 (0.57) ^+^	3.56 (1.04)	3.36 (0.80) ^+^	3.11 (1.05)
Ankle-foot orthoses usage (n, (%)) Yes No	0 (0.00%) 8 (100.00%)	1 (11.11%) 8 (88.89%)	0 (0.00%) 8 (100.00%)	1 (11.11%) 8 (88.89%)
Walking aids usage (n, (%)) Yes No	1 (12.50%) 7 (0.00%)	5 (55.56%) 4 (44.44%)	3 (32.50%) 5 (67.50%)	3 (33.33%) 6 (66.67%)
Using MS medication Yes No	8 (100.00%) 0 (0.00%)	8 (88.89%) 1 (11.11%)	8 (100.00%) 0 (0.00%)	8 (88.89%) 1 (11.11%)
Perceived importance of medication use (mean, (SD))	9.75 (0.71)	9.89 (0.33)	9.63 (0.74)	10.00 (0.00)
Physical limitations affecting the ability to perform self-care (n, (%)) Problems with hand Problems with feet Vision loss Hearing loss	2 (25.00%) 5 (62.50%) 1 (12.50%) 1 (12.50%)	1 (11.11%) 3 (33.33%) 0 (0.00%) 0 (0.00%)	1 (12.50%) 4 (50.00%) 1 (12.50%) 0 (0.00%)	2 (22.22%) 4 (44.44%) 0 (0.00%) 1 (11.11%)

^+^: n = 7. Includes age, BMI, ethnicity, education level, MS duration, EDSS score, assistive devices usage, use of MS medication, and perceived importance of medication use (1–10 Likert scale: 1 = not important, 10 = very important).

**Table 3 bioengineering-12-00932-t003:** QUEST results (n = 17).

Questions	Mean (SD)	Min Score
Satisfaction with the dimensions of the assistive device	4.76 (0.44)	4
Satisfaction with the weight	4.88 (0.33)	4
Satisfaction with the ease of adjusting the device	4.29 (1.05)	2
Satisfaction with how safe and secure of device is	4.71 (0.59)	3
Satisfaction with the durability	4.47 (1.07)	2
Satisfaction with ease of use	4.82 (0.39)	4
Satisfaction with the comfort of the device	4.71 (0.59)	3
Satisfaction with how effective the device is	4.53 (0.80)	3
Total satisfaction	37.18 (3.89)	27

Responses were collected via the Quebec User Evaluation of Satisfaction with Assistive Technology (QUEST 2.0) and semi-structured interviews, assessing satisfaction, perceived usefulness, and usability challenges (e.g., device adjustability, durability, comfort). Scores range from 1 (not satisfied) to 5 (very satisfied) for the QUEST items.

**Table 4 bioengineering-12-00932-t004:** Participants’ feedback on feasibility (n = 16) based on post-intervention interviews.

	n
Willing to use FVT	15
Willing to purchase the device	9
Will not purchase the device	2
Willing to purchase the device after improvements (stronger intensity, clear evidence, appropriate cost)	3
Missed using the device	3
No pain during device use	13
Potential adverse events during device use (spasm or pain)	2
Device preference All the aspects Ease of use Flexibility Easy to use Simplicity Fun to use Lightweight	6 5 1 2 1 1 1

**Table 5 bioengineering-12-00932-t005:** Changes in spatiotemporal parameters (N = 16) for experimental participants.

Parameter: m = Meters; s = Second		All Participants Mean (Standard Deviation)	*p* Values	Estimated Effect Size
**Gait speed (m/s)**	pre	0.92 (0.30)	0.014	0.34
	post	0.98 (0.27)		
**Stride length (m)**	pre	1.05 (0.23)	0.004	0.44
	post	1.11 (0.19)		
**Stride width (m)**	pre	0.19 ± 0.04	0.613	0.02
	post	0.19 ± 0.03		
**Stride time (s)**	pre	1.20 (0.19)	0.76	0.01
	post	1.19 (0.23)		
**More involved cadence (steps/min)**	pre	101.01 (16.18)	0.53	0.03
	post	102.36 (18.26)		
**Less involved cadence (steps/min)**	pre	105.34 (15.53)	0.74	0.01
	post	105.83 (13.97)		
**More involved stance (s)**	pre	0.79 (0.15)	0.62	0.02
	post	0.78 (0.16)		
**Less involved stance (s)**	pre	0.81 (0.15)	0.63	0.02
	post	0.79 (0.18)		
**More involved swing time (s) ^a^**	pre	0.40 (0.05)	0.57	0.02
	post	0.41 (0.07)		
**Less involved swing time (s) ^a^**	pre	0.40 (0.06)	0.97	<0.01
	post	0.40 (0.05)		
**Double limb support (s) ^a^**	pre	0.39 (0.11)	0.77	0.01
	post	0.39 (0.12)		

^a^ N = 15 due to missed measures.

**Table 6 bioengineering-12-00932-t006:** Changes in kinematic parameters for experimental participants (N = 16).

Parameter: LI-Less Involved; MI-More Involved		All Participants Mean (Standard Deviation)	*p* Values	Estimated Effect Size
**LI peak dorsiflexion at heel-off (°) ^a^**	pre	21.09 (2.41)	0.89	0.001
	post	21.24 (4.33)		
**LI peak plantarflexion before heel strike (°) ^a^**	pre	−5.02 (6.26)	0.18	0.13
	post	−7.03 (6.36)		
**LI peak dorsiflexion before heel-off (°) ^a^**	pre	11.55 (3.51)	0.97	<0.01
	post	11.52 (2.88)		
**LI peak plantarflexion during pre-swing (°) ^a^**	pre	−6.48 (7.30)	0.28	0.08
	post	−8.42 (7.38)		
**MI peak dorsiflexion at heel-off (°) ^a^**	pre	19.41 (2.96)	0.59	0.02
	post	19.79 (2.94)		
**MI peak plantarflexion before heel strike (°) ^a^**	pre	−7.02 (8.38)	0.056	0.24
	post	−8.99 (8.07)		
**MI peak dorsiflexion before heel-off (°) ^a^**	pre	10.60 (4.19)	0.62	0.02
	post	10.24 (4.97)		
**MI peak plantarflexion during pre-swing (°) ^a^**	pre	−9.61 (9.87)	0.043	0.26
	post	−11.88 (9.69)		
**LI knee flexion at heel stike (°)**	pre	8.75 (9.93)	0.41	0.05
	post	10.10 (8.65)		
**LI maximum knee flexion at stance (°)**	pre	43.68 (8.91)	0.86	0.002
	post	43.26 (5.53)		
**LI minum knee flexion at stance (°)**	pre	4.58 (7.82)	0.51	0.03
	post	5.61 (7.23)		
**LI maximum knee flexion at swing (°)**	pre	54.36 (11.26)	0.53	0.03
	post	55.80 (6.44)		
**LI minimum knee flexion at swing (°)**	pre	6.44 (10.75)	0.50	0.03
	post	7.59 (9.83)		
**MI knee flexion at heel stike (°)**	pre	8.12 (9.59)	0.44	0.04
	post	9.52 (8.23)		
**MI maximum knee flexion at stance (°)**	pre	40.62 (7.73)	0.31	0.07
	post	42.82 (4.68)		
**MI minimum knee flexion at stance (°)**	pre	2.39 (7.02)	0.41	0.05
	post	3.67 (4.83)		
**MI maximum knee flexion at swing (°)**	pre	52.46 (10.78)	0.31	0.07
	post	54.70 (8.03)		
**MI minimum knee flexion at swing (°)**	pre	5.89 (10.53)	0.80	0.004
	post	6.32 (8.87)		
**LI peak hip flexion at stance (°)**	pre	23.63 (10.55)	0.18	0.12
	post	28.12 (8.51)		
**LI peak hip extension at stance (°)**	pre	−13.73 (11.94)	0.36	0.06
	post	−10.90 (10.11)		
**LI peak hip flexion at swing (°)**	pre	24.75 (11.03)	0.16	0.13
	post	29.64 (8.46)		
**LI peak hip extension at swing (°)**	pre	−2.44 (12.86)	0.49	0.05
	post	−0.08 (10.02)		
**MI peak hip flexion at stance (°)**	pre	21.69 (12.09)	0.08	0.19
	post	27.41 (9.18)		
**MI peak hip extension at stance (°)**	pre	−14.86 (12.78)	0.29	0.07
	post	−11.66 (9.84)		
**MI peak hip flexion at swing (°)**	pre	23.20 (12.99)	0.11	0.16
	post	28.60 (8.91)		
**MI peak hip extension at swing (°)**	pre	−4.03 (13.97)	0.32	0.07
	post	−0.34 (10.64)		
**LI minimum knee flexion at swing (°)**	pre	6.44 (10.75)	0.50	0.03
	post	7.59 (9.83)		

^a^ N = 15 due to missed ankle measurements.

**Table 7 bioengineering-12-00932-t007:** Changes in kinetic parameters for experimental participants (N = 13).

Parameter: LI-Less Involved; MI-More Involved		All Participants Mean (Standard Deviation)	*p* Values	Estimated Effect Size
**LI peak plantar flexor moment (N · m/kg)**	pre	1.10 (0.20)	0.15	0.16
	post	1.15 (0.16)		
**LI peak ankle power in late stance (W/kg)**	pre	1.89 (0.77)	0.058	0.27
	post	2.13 (0.82)		
**LI peak ankle power in early stance (W/kg)**	pre	−0.84 (0.41)	0.74	0.01
	post	−0.86 (0.31)		
**MI peak plantar flexor moment (N · m/kg)**	pre	1.10 (0.26)	0.37	0.07
	post	1.14 (0.22)		
**MI peak ankle power in late stance (W/kg)**	pre	1.81 (0.95)	0.086	0.23
	post	2.03 (1.11)		
**MI peak ankle power in early stance (W/kg)**	pre	−0.71 (0.27)	0.077	0.24
	post	−0.79 (0.32)		
**LI peak knee extensor moment in early stance (N · m/kg)**	pre	0.85 (0.29)	0.88	0.002
	post	0.85 (0.21)		
**LI peak knee flexor moment in mid-stance (N · m/kg)**	pre	−0.17 (0.08)	0.57	0.03
	post	−0.18 (0.06)		
**LI peak knee flexor moment in early swing (N · m/kg)**	pre	0.11 (0.06)	0.86	0.003
	post	0.11 (0.04)		
**LI peak knee extensor moment in late swing (N · m/kg)**	pre	−0.28 (0.09)	0.19	0.14
	post	−0.29 (0.08)		
**LI peak knee power generation in stance (W/kg)**	pre	0.55 (0.40)	0.13	0.18
	post	0.71 (0.47)		
**LI peak knee power absorption in stance (W/kg)**	pre	−1.38 (0.65)	0.69	0.01
	post	−1.43 (0.49)		
**LI peak knee power absorption in swing (W/kg)**	pre	−0.97 (0.55)	0.55	0.03
	post	−1.01 (0.52)		
**MI peak knee extensor moment in early stance (N · m/kg)**	pre	0.74 (0.28)	0.19	0.14
	post	0.81 (0.27)		
**MI peak knee flexor moment in mid-stance (N · m/kg)**	pre	−0.18 (0.07)	0.67	0.02
	post	−0.19 (0.08)		
**MI peak knee flexor moment in early swing (N · m/kg)**	pre	0.08 (0.05)	0.076	0.24
	post	0.09 (0.06)		
**MI peak knee extensor moment in late swing (N · m/kg)**	pre	−0.26 (0.09)	0.27	0.10
	post	−0.28 (0.10)		
**MI peak knee power generation in stance (W/kg)**	pre	0.50 (0.29)	0.28	0.10
	post	0.57 (0.37)		
**MI peak knee power absorption in stance (W/kg)**	pre	−1.17 (0.58)	0.18	0.15
	post	−1.39 (0.61)		
**MI peak knee power absorption in swing (W/kg)**	pre	−0.87 (0.45)	0.065	0.26
	post	−1.03 (0.48)		
**LI peak hip flexion moment in late stance (N · m/kg)**	pre	1.19 (0.31)	0.31	0.09
	post	1.10 (0.33)		
**LI peak hip extension moment in stance (N · m/kg)**	pre	−0.33 (0.16)	0.26	0.10
	post	−0.37 (0.11)		
**LI peak hip flexor moment in early swing (N · m/kg)**	pre	0.21 (0.08)	0.25	0.11
	post	0.30 (0.25)		
**LI peak hip power generation in terminal stance (W/kg)**	pre	1.05 (0.36)	0.57	0.03
	post	1.09 (0.46)		
**LI peak hip power absorption in stance (W/kg)**	pre	−1.68 (0.78)	0.21	0.13
	post	−1.42 (0.56)		
**LI peak hip power generation in swing (W/kg)**	pre	0.61 (0.32)	0.61	0.02
	post	0.66 (0.30)		
**MI peak hip flexion moment in late stance (N · m/kg)**	pre	1.13 (0.38)	0.52	0.04
	post	1.07 (0.38)		
**MI peak hip extension moment in stance (N · m/kg)**	pre	−0.40 (0.17)	0.27	0.10
	post	−0.42 (0.16)		
**MI peak hip flexor moment in early swing (N · m/kg)**	pre	0.20 (0.09)	0.86	0.003
	post	0.20 (0.09)		
**MI peak hip power generation in terminal stance (W/kg)**	pre	1.00 (0.48)	0.67	0.02
	post	1.03 (0.38)		
**MI peak hip power absorption in stance (W/kg)**	pre	−1.54 (0.76)	0.33	0.08
	post	−1.36 (0.56)		
**MI peak hip power generation in swing (W/kg)**	pre	0.53 (0.30)	0.91	0.001
	post	0.54 (0.24)		

## Data Availability

The raw data supporting the conclusions of this article will be made available by the author upon request.

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
