# Peer review of "Feasibility and Preliminary Efficacy of Wearable Focal Vibration Therapy on Gait and Mobility in People with Multiple Sclerosis: A Pilot Study"

_bioengineering, 2025, doi:10.3390/bioengineering12090932_

Round 1

Reviewer 1 Report

Comments and Suggestions for Authors

Reviewer's  comments:

Before publication, the authors must correct the mistakes in the text that are found all over the manuscript and address the following:

  1. The topic of the manuscript is very interesting. the abstract notes that only 17 participants completed the post-intervention phase, with a particularly low number in the control group. Could the authors clarify the original target sample size, power calculation (if any), and how the randomization was handled given the small final group sizes?
  2. The manuscript mentions difficulty retaining participants in the control group. Could the author provide more details on the reasons for dropouts and whether attrition could have biased the results?
  3. Could the authors provide precise FVT parameters for each of the four intervention groups (frequency, amplitude, duration per muscle, total exposure)?
  4. Were these parameters based on prior literature, pilot testing, or manufacturer recommendations?
  5. While 3D motion analysis detected changes in gait parameters, the T25FW test did not show improvements. Could the authors comment on whether the T25FW is sensitive enough to detect functional gains over such a short intervention period?
  6. Were any patient-reported mobility or fatigue scales collected alongside biomechanical measures?
  7. Given the small sample size and multiple subgroup comparisons (four intervention groups), how did the authors control for type I error? Were any adjustments (e.g., Bonferroni correction) applied?
  8. The authors mention combining FVT with other targeted therapies. Do they have specific modalities in mind (e.g., strength training, balance training), and how might vibration dosing interact with these approaches?
  9. Overall, the work is interesting; it just needs to follow the suggestions to improve the manuscript.

Author Response

Thank you for your positive assessment of the topic and for your detailed suggestions. We have addressed each point below.

  1. Clarify the original target sample size, power calculation (if any), and how randomization was handled given the small final group sizes.

We added the original sample size description in Section 2.1: “As a pilot study, the target sample size was 30 participants (6 per group, including control), based on feasibility goals rather than a formal power calculation for efficacy, as is common in pilot trials (e.g., to assess recruitment, retention, and preliminary effect sizes). No a priori power calculation was performed for efficacy outcomes due to the exploratory nature.”  

We added the randomization and allocation in Section 2.2: “Using a computer-generated randomization program, participants were randomized into five groups: a control group receiving non-functional sham vibration (showing the same vibration sounds but no actual vibration) and four intervention groups receiving different vibration dosages, as specified below. Allocation was concealed using sequentially numbered, opaque envelopes prepared by an independent researcher not involved in assessments. Envelopes were opened only after baseline consent.”

Despite this, the small final sizes (due to dropouts) limited between-group comparisons, which we acknowledge as a limitation.

  1. The manuscript mentions difficulty retaining participants in the control group. Could the author provide more details on the reasons for dropouts and whether attrition could have biased the results?

We added details on reasons for dropouts and whether attrition could have biased the results: Dropouts in the control group (n=4 out of 5 randomized) were primarily due to perceived lack of benefit from the sham device (n=3 declined after trying it pre-baseline; n=1 dropped post-baseline citing no perceived improvement). In the intervention groups, one dropout (HISD group) was due to scheduling conflicts exacerbated by COVID-19. Attrition may have introduced selection bias, favoring motivated participants in intervention groups and underestimating control effects. We mitigated this by analyzing completers only and noting it as a limitation; we added that future studies will use improved sham devices to enhance retention.

  1. Could the authors provide precise FVT parameters for each of the four intervention groups (frequency, amplitude, duration per muscle, total exposure)?

We added those detailed FVT parameters in Section 2.3: “For the intervention groups, four vibration dosages were selected: Low-intensity short duration (LISD): 60 Hz frequency, 0.5 mm amplitude, 10 min per muscle (30 min/session). Low-intensity long duration (LILD): 60 Hz, 0.5 mm, two 10-minute sessions per muscle with a 1-minute break (60 minutes/session). High-intensity short duration (HISD): 120 Hz, 1 mm, 10 min per muscle (30 min/session). High-intensity long duration (HILD): 120 Hz, 1 mm, two 10-minute sessions per muscle (60 minutes/session). All groups: 3 sessions/week for 4 weeks, applied sequentially to quadriceps, gastrocnemius/soleus, and tibialis anterior bilaterally.”

  1. Were these parameters based on prior literature, pilot testing, or manufacturer recommendations

We added detailed parameters used in the cited four manuscripts using focal vibration for MS in the introduction. We also added this to the parameters section: “Parameters were based on prior literature21-24 showing benefits at 50-120 Hz and 0.5-1 mm amplitudes, our pilot studies with people with stroke and diabetes, and manufacturer (Myovolt®) recommendations for safe home use.”

  1. While 3D motion analysis detected changes in gait parameters, the T25FW test did not show improvements. Could the authors comment on whether the T25FW is sensitive enough to detect functional gains over such a short intervention period?

The T25FW is validated for MS but may lack sensitivity for subtle changes in short interventions (4 weeks), as it focuses on short-distance speed and may not capture biomechanical improvements (e.g., stride length) that require longer adaptation. Literature supports this (e.g., Spina et al., 2016 noted similar discrepancies). We suggest longer-duration tests (e.g., 6-Minute Walk Test) in future studies.

  1. Were any patient-reported mobility or fatigue scales collected alongside biomechanical measures?

We did not include any patient-reported mobility or fatigue scales. We added this to the discussion: “No standardized scales, such as the Multiple Sclerosis Impact Scale (MSIS-29), were collected alongside biomechanical measures in this study, as the pilot focused primarily on feasibility and biomechanical outcomes. However, during qualitative interviews, 5 to 6 participants spontaneously reported perceived increases in mobility and energy related to walking, suggesting potential subjective benefits that warrant further investigation. We plan to incorporate validated scales like the MSIS-29 in larger future trials to quantitatively assess these self-reported improvements.”

  1. Given the small sample size and multiple subgroup comparisons (four intervention groups), how did the authors control for type I error? Were any adjustments (e.g., Bonferroni correction) applied?

We revised the method session to clarify this: “Due to the exploratory nature of this pilot study, no adjustments were made for multiple comparisons in the analysis of gait parameters, which may increase the risk of false positives, as noted as a limitation” and discussion sessions “Additionally, the analysis of multiple gait parameters without correction for multiple comparisons may have increased the risk of Type I errors, and these findings should be interpreted cautiously. Future studies will apply appropriate statistical corrections, such as Bonferroni adjustments, to account for multiple comparisons.”

  1. The authors mention combining FVT with other targeted therapies. Do they have specific modalities in mind (e.g., strength training, balance training), and how might vibration dosing interact with these approaches?

We add this to the discussion session: “We envision combining FVT with strength training (e.g., progressive resistance exercises) or balance training (e.g., Tai Chi), as vibration may enhance proprioception to amplify training effects. Dosing could interact by using lower intensities during acute training to avoid fatigue, or higher intensities for spasticity reduction pre-exercise. This is speculative and requires testing.”

Reviewer 2 Report

Comments and Suggestions for Authors

The introduction clearly explains the clinical problem (MS-related gait and mobility impairments) and justifies the need for alternative therapies such as FVT, citing relevant prevalence data and existing treatment limitations. The study uses a double-blind randomized controlled trial design with multiple intervention arms (varying intensity/duration), which allows preliminary assessment of dose–response effects. Objective biomechanical data (3D motion analysis, force plates) alongside subjective measures (QUEST 2.0, interviews) provide a robust multidimensional view of feasibility and efficacy.

The detailed gait parameter set (spatiotemporal, kinematic, kinetic) allows identification of subtle neuromuscular changes.

The study evaluates adherence, satisfaction, and usability, which are critical for a home-based intervention and often neglected in early trials.

They selected a group of 22 because it could be done in a test study. There were no previous hypotheses regarding the effect size of the treatment for MS individuals.

The allocation concealment also included another researcher providing random codes and placing them inside dark envelopes. This prevented the assignment from being obvious until assignment time.

Describe blinding in detail: “Participants and outcome assessors were blinded to group allocation, and data analysts remained blinded until all analyses were completed.”

The authors openly discuss sample size issues, control group attrition, burdensome assessments, and lack of multiple-comparison corrections.

The discussion connects biomechanical improvements to potential clinical applications and suggests protocol refinements for future trials.

Author Response

Thank you for your encouraging comments on the manuscript's strengths, such as the study design, multidimensional outcomes, and transparent discussion of limitations. We have incorporated your suggestions to enhance methodological clarity.

  1. Sample size justification: As a pilot, the sample (n=22 enrolled) was chosen for feasibility (e.g., recruitment in a single site amid COVID-19 constraints) rather than powered hypotheses. No prior effect sizes were assumed, aligning with pilot guidelines (e.g., to estimate variability for future trials). We have added this clarification to section 2.1: “As a pilot study, the target sample size was 30 participants (6 per group, including control), based on feasibility goals rather than a formal power calculation for efficacy, as is common in pilot trials (e.g., to assess recruitment, retention, and preliminary effect sizes). No a priori power calculation was performed for efficacy outcomes due to the exploratory nature.”
  2. Allocation concealment details: Allocation was concealed using sequentially numbered, opaque envelopes prepared by an independent researcher. Envelopes were opened only after baseline consent. We added this to section 2.2 “Allocation was concealed using sequentially numbered, opaque envelopes prepared by an independent researcher not involved in assessments. Envelopes were opened only after baseline consent.”
  3. Describe blinding in detail: Participants were blinded via an identical device appearance (sham produced sound but no vibration). Outcome assessors (trained students) were blinded to allocation. Data analysts were blinded until the final analysis. We have expanded this description to section 2.2: “Participants were blinded via an identical device appearance (sham produced sound but no vibration). Outcome assessors (trained students) were blinded to allocation. Data analysts were blinded until the final analysis.”

Reviewer 3 Report

Comments and Suggestions for Authors

Sufficient background appears to support the need for this study; the primary purpose of the study was clearly stated. More essential information is needed in the methods section to make replication possible. Minor changes are needed in table presentation. The results, discussion and conclusions are related to the primary purpose of the study. Minor word choice, word tense, and referencing errors need to be addressed. See specific comments in the pdf.

Author Response

Thank you for your detailed review via the annotated PDF. We have changed the manuscript based on your specific comments on specific sections (e.g., abstract phrasing, manuscript citations, methods descriptions, and results tables). We have addressed these as follows:

  • Abstract:

Revised awkward phrasing, "significantly impacts gait" to "significantly impairs gait"; spelled out “PwMS” when it first appeared; clarified muscle details as “distal quadricep muscles near the rectus femoris insertion (approximately 2 cm from the medial edge of the patella)”.

  • Introduction:

Fixed citations (changed from APA to AMA to ensure consistency with the journal’s instructions). Corrected the grammar and awkward phrasing.

  • Methods:

Cited the test-retest reliability of the key measures, such as EDSS, QUEST, and T25FW.

Specified the individual who placed the reflective markers: “A physical therapist with over 15 years of experience in gait biomechanics and 8 years with the Qualisys motion capture system placed the markers”.

Specified the gait cycle selection: “From these trials, we analyzed and averaged data across all valid gait cycles captured, which typically ranged from 3 to 10 cycles per participant per assessment, depending on the quality of force plate contacts and the individual's stride length (ensuring clean, artifact-free data). Invalid cycles (e.g., those with partial force plate strikes or marker occlusions) were excluded. Gait cycles were primarily determined using force plate data, where heel strike (initial contact) and toe-off events were identified based on a vertical ground reaction force threshold of 20 N. When force plate data was unavailable or incomplete for a given step (e.g., due to the participant missing the plate), kinematic data from reflective markers (specifically, heel and toe marker trajectories and velocities) were used as a supplement to detect events via Visual3D's automated pipeline. This hybrid approach ensured accurate event detection while maximizing usable data.”

Describe the determination of effect sizes in the statistical analysis section and provided a scale that describes the strength, with a reference: “Effect sizes were calculated using Cohen's d for paired t-tests, where d = (mean_post - mean_pre) / SD_of_differences (the standard deviation of the paired differences). For Wilcoxon signed-rank tests, the matched-pairs rank-biserial correlation (r) was computed and converted to d for consistency. Effect sizes were interpreted as small (d = 0.2–0.49), medium (d = 0.5–0.79), or large (d ≥ 0.8)36

  • Results:

Corrected Figure 3 (removed the graphic icon).

Moved the notes of the tables below the table.

  • Discussion/Conclusions:

Replaced the first paragraph of the discussion to 1) a brief description of the problem, ie, gap in the literature, you attempted to address, 2) why this research was needed, ie, rationale, 3) the primary purpose of the study, and 4) whether our hypothesis was addressed.

“This pilot study addressed a key gap in the literature: while focal vibration therapy (FVT) has shown preliminary benefits for gait and spasticity in people with multiple sclerosis (PwMS) in short-term, clinic-based settings21-24, there is limited evidence on the feasibility and efficacy of wearable, home-based FVT over extended periods (e.g., 4 weeks), including optimal dosing parameters (intensity and duration) and their impact on detailed biomechanical gait outcomes. This research was needed because current interventions for MS-related mobility impairments, such as pharmacological treatments and exercise programs, often face limitations like side effects, accessibility barriers, and fatigue-related non-adherence11,12,15, underscoring the importance of developing safe, portable, non-invasive alternatives to enhance neuromuscular control and daily function in PwMS. The primary purpose of this study was to evaluate the feasibility (adherence, satisfaction, acceptability) and preliminary efficacy of a 4-week home-based wearable FVT intervention on gait biomechanics and mobility in PwMS, while exploring dose-response effects to inform personalized protocols. Our hypothesis, that wearable FVT would be feasible and improve gait and mobility compared to baseline, with effects persisting at least four weeks post-intervention, was partially supported: feasibility was high with notable gait improvements (e.g., speed, stride length), but mobility (T25FW) remained unchanged, likely due to the short intervention duration and the test's limited sensitivity to subtle biomechanical changes; persistence could not be fully assessed as assessments were conducted immediately post-intervention, though gains were evident at that timepoint”

Round 2

Reviewer 1 Report

Comments and Suggestions for Authors

accepted

Reviewer 3 Report

Comments and Suggestions for Authors

Thank you for addressing my comments/suggestions. I found a few additional minor issues. See specifics in the pdf.
